https://doi.org/10.1038/s41467-019-10498-1　　**OPEN**

# Neuronal cell-subtype specificity of neural synchronization in mouse primary visual cortex

Ulf Knoblich[1,5], Lawrence Huang[1,5], Hongkui Zeng [1] & Lu Li[2,3,4]

Spatiotemporally synchronised neuronal activity is central to sensation, motion and cognition. Brain circuits consist of dynamically interconnected neuronal cell-types, thus elucidating how neuron types synergise within the network is key to understand the neuronal orchestra. Here we show that in neocortex neuron-network coupling is neuronal cell-subtype specific. Employing in vivo two-photon (2-p) Calcium (Ca) imaging and 2-p targeted whole-cell recordings, we cell-type specifically investigated the coupling profiles of genetically defined neuron populations in superficial layers (L) of mouse primary visual cortex (V1). Our data reveal novel subtlety of neuron-network coupling in inhibitory interneurons (INs). Parvalbumin (PV)- and Vasoactive intestinal peptide (VIP)-expressing INs exhibit skewed distributions towards strong network-coupling; in Somatostatin (SST)-expressing INs, however, two physiological subpopulations are identified with distinct neuron-network coupling profiles, providing direct evidence for subtype specificity. Our results thus add novel functional granularity to neuronal cell-typing, and provided insights critical to simplifying/understanding neural dynamics.

[1] Allen Institute for Brain Science, 615 Westlake Ave N, Seattle, WA 98109, USA. [2] Guangdong Provincial Key Laboratory of Malignant Tumor Epigenetics and Gene Regulation, Sun Yat-sen Memorial Hospital, Sun Yat-sen University, 107 Riverside West Road, 510120 Guangzhou, Guangdong, China. [3] Medical Research Centre, Sun Yat-sen Memorial Hospital, Sun Yat-sen University, 107 Riverside West Road, 510120 Guangzhou, Guangdong, China. [4] Centre for Brain Science and Brain-Inspired Intelligence, Guangdong-Hongkong-Macao Greater Bay Area, Pearl River Delta, China. [5]These authors contributed equally: Ulf Knoblich, Lawrence Huang. Correspondence and requests for materials should be addressed to L.L. (email: lilu67@mail.sysu.edu.cn)

The fundamental question in neuroscience is how the brain dynamically acquires, processes, stores and retrieves information to adapt the animal to rapidly changing environment. To perform the required neural computation, brain neurons coordinate or synchronise with their neighbours, in a way analogous to musicians play symphonies together with peer instrumentalists. Such synchronised neuronal activity is collectively reflected by brain oscillations observed in electroencephalogram (EEG), local field potential (LFP), extracellular spiking and intracellular membrane potential (Vm) recordings of neural dynamics at multiple spatiotemporal scales[1–8]. Synchronised neuronal activity may play key roles in emergent functions of neural networks and has been found critical to brain states, perception, learning and other higher-order cognitive functions including attention[1–8], but mechanistic knowledge of neuronal synchronisation remains limited, primarily due to the overwhelming complexity of brain networks.

Structurally brain networks can be simplified by neuronal cell-typing, which categorises neurons into finite distinct groups with characteristic morphology, physiology, connectivity and genetics[9–12]. Considering the structure-function correspondence, synchronised neuronal activity that reverberates on brain networks could also be simplified by examining neuronal cell-type specific within- and between-type/subtype interactions[8,13]. In neocortex, the distribution of anatomical connectivity strength between neighbouring excitatory pyramidal (Pyr) neurons has been found matching with their sensory response similarity[14–16], validating the structure-function correspondence at the single-neuron level. Efforts have also been made to explore the cell-type dependence in neuronal synchronisation. For example, diversified neuron-network coupling has been found in Pyr neurons, but not PV neurons, that Pyr cells may couple their spikes strongly with the LFP thus are choristers, or care little about synchronising with their neighbours so behave like soloists[8,17]. Although above pieces of evidence suggest that neural dynamics could be explained, at least partially in Pyr neurons, by inter-neuron correlativity, whether and to what extent these observations generalise to other neuronal cell-types still remains largely unknown. Given the heterogeneity in anatomical connectivity[18,19], a parsimonious hypothesis suggests cell-type/subtype specificity in neuronal synchronisation. However, this possibility hasn't been fully investigated.

We systematically tested this hypothesis by performing in vivo 2-p Ca imaging in L2/3 of mouse V1 to compare the pair-wise, within-type correlation profiles of excitatory and three major inhibitory neuron types (Figs. 1 and 2, see Methods). We focused on superficial neurons, because they integrate information from bottom-up and top-down pathways, as well as local recurrent inputs to bias animals' behaviour[20,21] and are readily accessible to 2-p microscopy. Taking advantage of our newly developed transgenic mouse lines[22,23] that cell-type specifically express genetically encoded Ca indicators[24] (GECIs, including GCaMP6s and GCaMP6f), we simultaneously imaged local neuron populations within $400 \times 400\,\mu m$ fields of view (FOVs) at various depths in awake, head-fixed mice with single-neuron resolution (Fig. 1a–c and Supplementary Fig. 1).

## Results

**Novel cell-subtype specificity in neuronal synchronisation.** We first examined Pyr neurons by analysing the pairwise ($n = 6936$ pairs) zero-time lag correlation coefficients (CCs) of fluorescence (F) changes ($\Delta F/F$) between individual fluorescently active regions of interest ($n = 497$ of 530 ROIs in 18 FOVs) in *Emx1-IRES-Cre;CaMK2a-tTA;Ai94* mice (expressing GCaMP6s in pan-excitatory cortical neurons, $n = 3$). Due to reported heterogeneity

of visual responsiveness among neuron types[25–27], we focused on spontaneous activity. As exemplified in Fig. 1d, within-type correlation was quantified for each active Pyr cell by the percentage of its correlated Pyr neighbours per FOV. Overall, we found Pyr neurons showed a diverse correlation profile (Figs. 1d and 2a), resembling the broad and continuous distribution from weakly network-coupled soloists to strongly coupled choristers reported previously[8,17]. Across all mice, the distribution of CC between active Pyr pairs was highly skewed that only a small fraction of Pyr neurons were strongly correlated[8,16] (Fig. 2e). No apparent fine-scale clustering was found in Pyr neurons, because spatially the correlation decreased monotonically along the distance, confirming the salt and pepper topology[15,16,28] of rodent V1 (Fig. 2i, Supplementary Fig. 1). All these above resulted in a generally weak correlation in Pyr neurons[14–16] (median: 0.14; mean ± s.e.m.: 0.14 ± 0.02, Fig. 2k). In summary, our imaging results were highly consistent with previous studies[14–16,28], validating our Ca imaging approach.

Next we tested whether, and to what extent, these findings in Pyr neurons generalised for IN types. In mouse V1 INs comprise ~20% of the entire neuron population and can be categorised into three non-overlapping types[29], namely PV-, SST- and 5HT3aR-expressing INs (VIP is a sub-group of 5HT3a INs). Due to the sparse distribution of cortical INs, we chose to perform 2-p Ca imaging in *Pvalb-IRES-Cre;Ai163* (GCaMP6s), *VIP-* and *SST-IRES-Cre;Ai14;Ai148* (GCaMP6f) mice[23], which all co-expressed GECIs with a red cytosol fluorescence protein tdTomato (tdT). For PV INs, we imaged 252 fluorescently active PV INs (total 355 ROIs from 18 FOVs) in awake, head-fixed *Pvalb-IRES-Cre;Ai163* mice ($n = 3$) and analysed the pairwise ($n = 1864$ pairs) within-type CCs. Compared with Pyr neurons, PV INs were highly active and homogeneously synchronised[8,17] (Figs. 1e and 2b–k). Accordingly their CC distribution showed a significant rightward shift towards strong synchronisation ($p < 0.001$, two-sample Kolmogorov–Smirnov test, Fig. 2a–k), and the average CC (median: 0.53; mean ± s.e.m.: 0.56 ± 0.03, $n = 3$) was significantly higher than Pyr neurons ($p < 0.001$, $t$-test, Fig. 2k). Unlike Pyr neurons, correlation between PV INs remained strong within 200 µm before decreasing (Fig. 2i), indicating the highly synchronising behaviour of PV INs may relate to their global role in brain oscillations[8,17,30,31].

For VIP INs, Ca imaging data showed that they behaved similarly to PV INs (Figs. 1f and 2c–k). We imaged 333 fluorescently active VIP INs (total 368 ROIs from 16 FOVs) in awake, head-fixed *VIP-IRES-Cre;Ai14;Ai148* mice ($n = 3$). The CC profile from 3420 pairs of active VIP INs demonstrated they were also highly synchronised compared with Pyr neurons (Fig. 2c–k): CC distribution rightward shifted significantly ($p < 0.001$, two-sample Kolmogorov–Smirnov test), and the average CC value (median: 0.29; mean ± s.e.m.: 0.28 ± 0.02, $n = 3$) was significantly higher than Pyr neurons ($p < 0.01$, $t$-test).

**Two distinct subtypes identified in SST INs.** To our surprise, for SST INs our imaging data identified 2 SST subpopulations or phenotypes (Fig. 1g). We investigated the within-type correlation ($n = 822$ pairs) of fluorescently active L2/3 SST INs ($n = 184$ of 342 ROIs imaged from 20 FOVs) in awake, head-fixed *SST-IRES-Cre;Ai14;Ai148* mice ($n = 4$). Forty-two percent (42%, 77 of 184, $n = 156$ within-subtype pairs, Fig. 2d) of SST INs (termed Subtype I for simplicity) were spontaneously active, but uncorrelated to almost all (≥95%) active SST neighbours[32] (CC median: 0.03; mean ± s.e.m.: 0.02 ± 0.01, $n = 4$). The rest SST INs (58%, 107 of 184, $n = 341$ within-subtype pairs), however, had intermediate magnitude of CC between Pyr and PV neurons (median: 0.32; mean ± s.e.m.: 0.31 ± 0.03, $n = 4$, Figs. 1g and 2d–k), ruling out

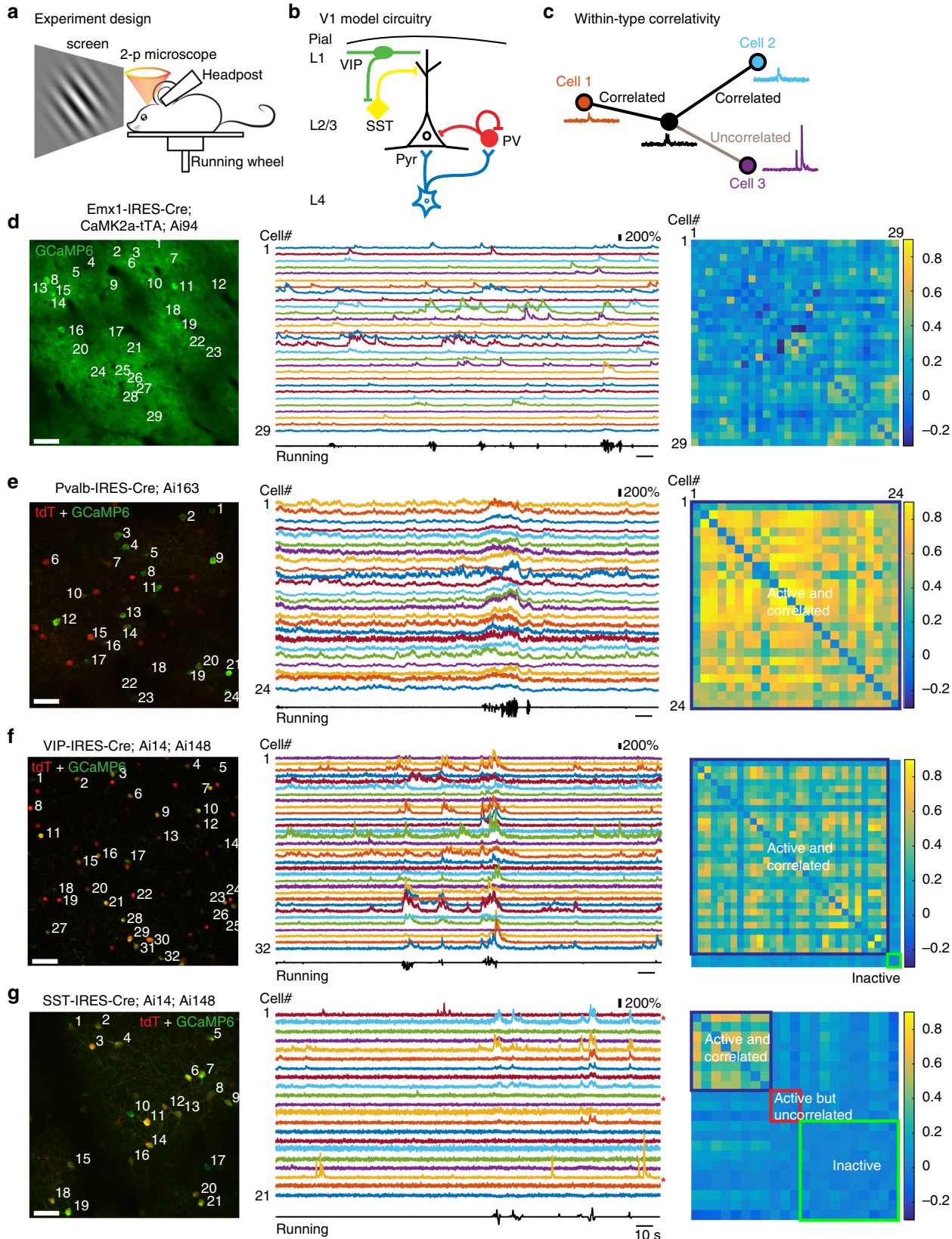

the possibility of mislabeled PV neurons[33,34]. These SST INs were thus called Subtype II. The separation between SST subtype I and II was NOT caused by a sampling bias, because under the same condition the percentages of uncorrelated Pyr, PV and VIP cells were only 2, 0 and 2%, respectively. Consistently, CC distribution was long-tailed, supporting the existence of multiple SST subtypes. In addition, the spatial distribution also differed between Subtype I and II SST INs: Subtype I was more evenly scattered within the $x$–$y$ plane, but Subtype II tended to cluster more heavily so that their correlation strength rapidly dropped beyond

**Fig. 1** Novel neuronal cell-subtype specificity of neocortical synchronisation during wakefulness. **a–c** Experimental design. In vivo 2-p Ca imaging (**a**) was performed in awake, head-fixed transgenic mice to investigate excitatory and three major inhibitory neuron types (**b**) by characterising the pair-wise, within-type cross-correlation coefficient (CC) profiles (**c**) in L2/3 of mouse V1. **d** Example Ca imaging in an awake, head-fixed *Emx1-IRES-Cre;CaMK2a-tTA; Ai94* mouse showing spontaneous CC of V1 L2/3 Pyr neurons. Left: Z-projection (time series) of green (GCaMP6s) fluorescence images within the 400 × 400 μm field of view (FOV) at 162 μm underneath the pial surface, numbers indicate individual regions of interest (ROIs) analysed. Scale bar: 50 μm. Middle: ΔF/F traces of 250 s imaging of Pyr neurons as labeled in left panel. Running velocity was displayed at the bottom. Right: Within-type CC matrix of active Pyr neurons within the FOV, constructed from the CCs at zero-time lag of corresponding Pyr pairs. (**e–g**). Example Ca imaging of PV (24 ROIs at 280 μm), VIP (32 ROIs at 160 μm) and SST (21 ROIs at 150 μm) INs, respectively, to demonstrate the cell-subtype specificity. Left panels showed z-projected images of overlaid tdT (Red) and GECI (green) fluorescence signals, the rest were labeled as in **d**. Red asterisks (*) indicate fluorescently active, but network-uncoupled cells. Note that PV and VIP INs in V1 L2/3 were highly synchronised (**e, f**) but SST INs showed two subgroups/subtypes (**g**): One SST subgroup was spontaneously active but uncorrelated to any active SST neighbours (Subtype I, red box in CC matrix in **g**). Subtype II SST INs, however, were strongly network-coupled (dark blue box in CC matrix in **g**). For display purposes, CC matrices in **f, g** were sorted. Fluorescently inactive ROIs (green boxes in CC matrixes) were displayed here but not analysed (but see our electrophysiological data in Figs. 3–5). The mouse image in a was reproduced from Li et al. 2017 Nat Commun. 8:15604. https://doi.org/10.1038/ncomms15604 with permission

100 μm (Supplementary Fig. 2).Taken together, our Ca imaging data revealed novel SST subtype specificity in cortical synchronisation: L2/3 SST INs consisted of two functional subtypes largely differing in their within-type correlativity, indicating novel functional granularity.

**Selective modulation of SST Subtypes by locomotion.** Consistent with these findings, animals' arousal states, such as locomotion differentially modulated SST Subtype I and II neurons, supporting this novel subtype specificity. In mouse V1 it has been reported that locomotion upregulates visual responsiveness but leaves the orientation selectivity unaffected[35,36]. Cell-type dependent effects of locomotion on neuronal activity have also been described[37–39], but whether and how locomotion affect the correlation profile by neuron types are not completely understood. We quantified and compared within-type CC profiles of Pyr, PV, VIP and SST neurons by arousal states (i.e. stationary vs. locomotion) in qualified FOVs (Supplementary Fig. 3, also see Methods). Our data showed that locomotion modulation of CC varied considerably between neuron types: Pyr (stationary vs. locomotion, mean ± s.e.m.: $0.13 \pm 0.005$ vs. $0.31 \pm 0.01$, n = 162 ROIs, $p < 0.001$, t-test) and VIP ($0.24 \pm 0.007$ vs. $0.36 \pm 0.009$, n = 251 ROIs, $p < 0.001$, t-test) significantly increased their within-type CC during locomotion, but PV CC decreased significantly ($0.50 \pm 0.01$ vs. $0.42 \pm 0.01$, n = 141 ROIs, $p < 0.001$, t-test). Interestingly, for SST INs locomotion increased the within-type CC in almost all uncorrelated Subtype I SST cells (stationary vs. locomotion: $0.05 \pm 0.004$ vs. $0.20 \pm 0.02$, n = 20 ROIs, imaged from six FOVs in two animals, $p < 0.001$, t-test), but bidirectionally modulated the correlated Subtype II neurons and resulted in no total net effects ($0.19 \pm 0.01$ vs. $0.21 \pm 0.03$, n = 29, $p = 0.47$, t-test; Supplementary Fig. 3). These results suggested SST Subtype I INs were more sensitive to locomotion compared with SST Subtype II, which were consistent with previous finding that SST neurons may be more dependent on the top-down pathway and/or global brain states[32,40].

**Corroborating Ca imaging data with 2-p targeted patching.** Due to the slow dynamics of intracellular Ca activity, data with higher temporal resolution are required for further insights on cell-type specificity in synchronised neural activity, which usually occurs at the millisecond scale. It is also necessary to unbiasedly include low-spiking and/or spiking-but-fluorescently-inactive neurons[41], because 46% of SST INs we imaged were fluorescently inactive, compared with 6, 29 and 10% for Pyr, PV and VIP neurons, respectively, which may confound our interpretation of SST subtypes. Finally, it is critical to look into the oscillation patterns of underlying Vm for mechanistic explanations of

the subtype specificity of V1 L2/3 SST INs revealed by Ca imaging. In vivo 2-p targeted whole-cell recording is ideal to achieve these goals, because blind electrophysiological techniques such as extracellular recording are known to bias towards active neurons. Here we chose to conduct 2-p targeted whole-cell current-clamp recordings[42] of spontaneous and visually evoked Vm activities under anaesthesia, during which brain networks were highly synchronised[1–3] (Supplementary Fig. 4). This was because, as we reasoned that, if asynchronous brain network during wakefulness[5–8] contributes to the distinct behaviours of SST subtypes, minimising these effects would confirm the existence of innately uncorrelated Subtype I SST INs. Also, it was technically less challenging to perform in vivo whole-cell patching under anaesthesia, compared to that in awake animals.

We first targeted and whole-cell patched tdT+ Pyr neurons to quantify Vm oscillation features and the coupling strength of Vm to local network dynamics (reported by simultaneously recorded electrocoritcogram, ECoG, Fig. 3a and b, Supplementary Fig. 5, also see Methods). Consistent with previous studies[5–8,32], Vm data from tdT+ Pyr neurons (n = 15) across three mouse lines (eight neurons in *Rorb-IRES-Cre;Ai14*, five in *Cux2-CreERT2;Ai14* and two in *Scnn1a-IRES-Cre;Ai14* mice, respectively) confirmed that Pyr neurons were highly synchronised under anaesthesia (Fig. 3c–g, Supplementary Fig. 5). As shown in Fig. 3d, Vm in all Pyr neurons spontaneously oscillated between a hyperpolarised resting Down state (blank regions) and depolarised Up state (gray shaded regions), resulting in a skewed, bimodal Vm distribution[5,7,43] (Pearson's Coefficient of Skewness $\xi_M = -0.80 \pm 0.07$, mean ± s.e.m., same below, Fig. 3e and f). Spikes occurred sparsely during the Up state, faithfully corresponding with desynchronised phases of simultaneously recorded ECoG (Fig. 3d). Consequently, as well demonstrated in mice and monkeys[5,7], subthreshold Vm (spike removed, see Methods and Supplementary Fig. 5) coupled well with ECoG (Vm-ECoG CC = $-0.21 \pm 0.04$, n = 3), confirming Pyr neurons under anaesthesia receive large but intermittent volleys of excitatory inputs[5,7]. Brief depolarisation of Vm by current steps around the termination of the desynchronisation phase of ECoG was insufficient to override the coupling (Fig. 3d, but see ref. [44], proving a dominant network influence on Vm oscillations in Pyr neurons. Our data were highly consistent with the literature[5,7,8,17,32,43]. In summary, the skewed Vm distribution and tight Vm-ECoG coupling from the convergence of network excitation together demonstrated Pyr neurons were highly synchronised with the local network under anaesthesia.

**Vm mechanisms underlying distinct behaviours of SST INs.** To obtain mechanistic insights, we next target-patched tdT+ cells (n = 20) in *SST-IRES-Cre;Ai14* mice. Our whole-cell data

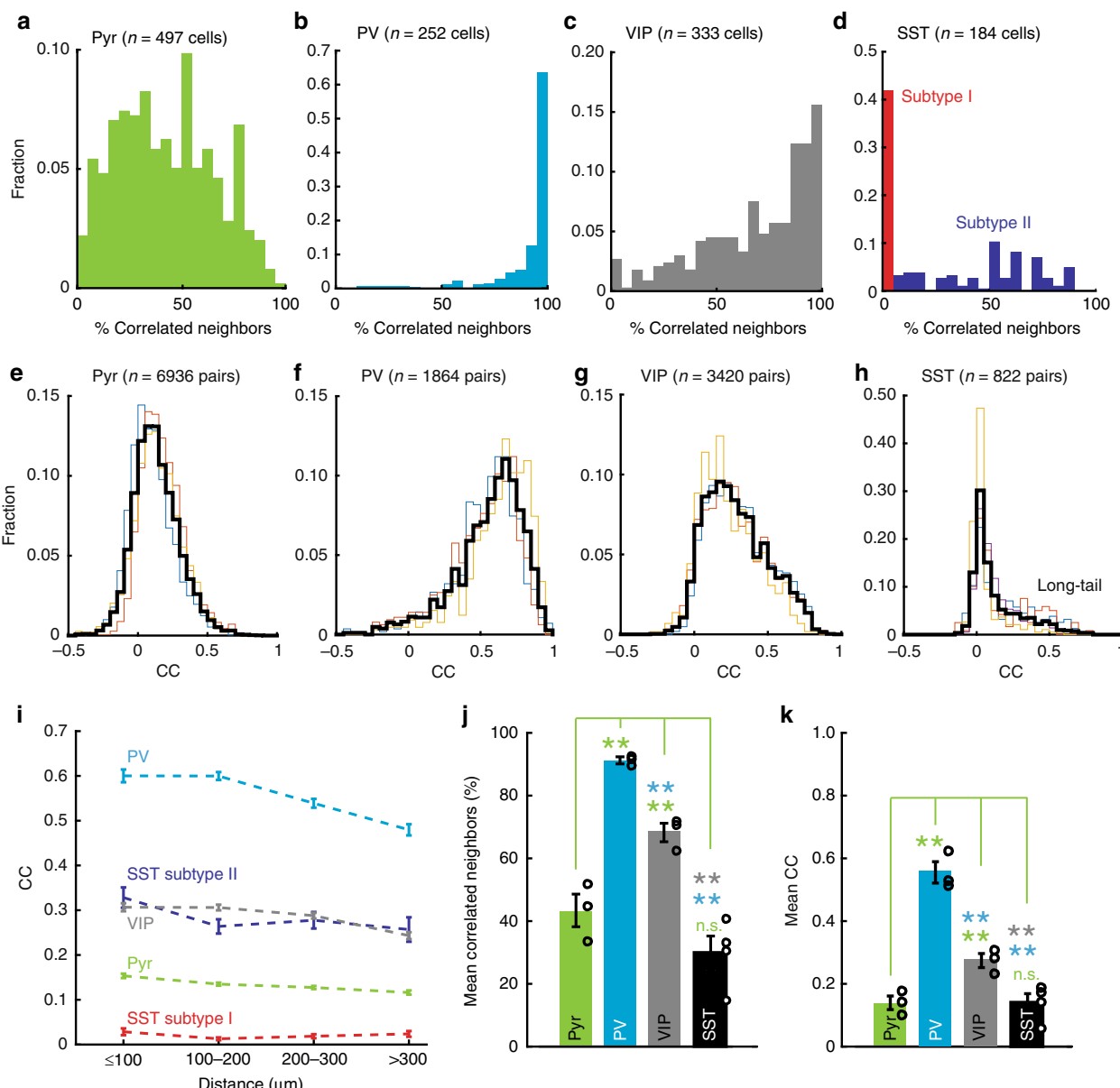

**Fig. 2** Population data of neuronal cell-subtype specific correlation. **a–d** Population histograms showing the within-type correlation distributions in Pyr (**a**), PV (**b**), VIP (**c**) and SST (**d**) neurons, respectively. Each histogram plotted the fraction of active cells against the percentage of correlated active neighbours (see Methods). Note the difference in CC distributions between neuron types: Pyr neurons (**a**) exhibited a broad and continuous distribution, showing a versatile correlation profile. PV INs (**b**) were spontaneously active and highly synchronised, almost all PV INs was strongly correlated with every other PV IN in the same FOV. VIP INs (**c**) were also highly synchronised with each other, significantly stronger than Pyr neurons but weaker than PV INs. SST INs (**d**), on the other hand, showed a subtype specificity. One subpopulation (red, Subtype I) were spontaneously active, but uncorrelated to nearly any SST neighbours. Subtype II SST cells (dark blue), however, were strongly coupled. **e–h**. CC distributions in Pyr (**e**), PV (**f**), VIP (**g**) and SST (**h**) neurons, respectively. Thin lines represented data from individual animals. Note the long-tailed distribution in SST INs, which was in line with the existence of two subtypes. **i** Relationships between CC strengths and corresponding spatial distance in Pyr (green), PV (light blue), VIP (gray) and SST subtype I (red) and II (dark blue) neurons, respectively (two-way ANOVA, $p < 0.001$). **j** Average % of correlated neighbours across animals in Pyr (green), PV (light blue), VIP (green) and SST (black) neurons, respectively. Statistical significance was colour-coded. **k** Average CC in Pyr (green), PV (light blue), VIP(green) and SST (black) neurons, respectively. **$p < 0.01$, n.s.: not significant. Error bar: s.e.m.

confirmed two SST subtypes (Figs. 4 and 5): 40% of SST INs (eight of 20, Fig. 4a–c) showed unimodally distributed Vm ($\xi_M = -0.19 \pm 0.08$, $n = 8$, $p < 0.01$, $t$-test), which was in stark contrast to the bimodal Vm distribution in Pyr neurons (Fig. 4c, e and h). As predicted by the unimodal Vm distribution, Vm-ECoG correlation in these SST INs was significantly weaker than Pyr neurons (CC = $-0.08 \pm 0.04$, $n = 4$, $p < 0.01$, $t$-test, Fig. 4f, g, i and j), suggesting little, if any, spontaneously occurring pure network excitation. Four of eight (50%) Subtype I SST

INs had much depolarised resting Vm (Vrest) at ~−40 mV, and were highly spontaneous active (Fig. 4c), similar to those reported in primary somatosensory cortex (S1) of awake, head-fixed mice[32]. However, the uni-modality of Vm distribution was independent of Vrest ($-52.6 \pm 3.9$ mV, $n = 8$), as demonstrated by the unimodal Vm distribution in spontaneously inactive Subtype I SST neurons with hyperpolarised Vrest between −50 and −70 mV ($n = 4$, Fig. 4h). Furthermore, driving Vm away from the membrane reversal potential (Vrev, ~−40 mV) by

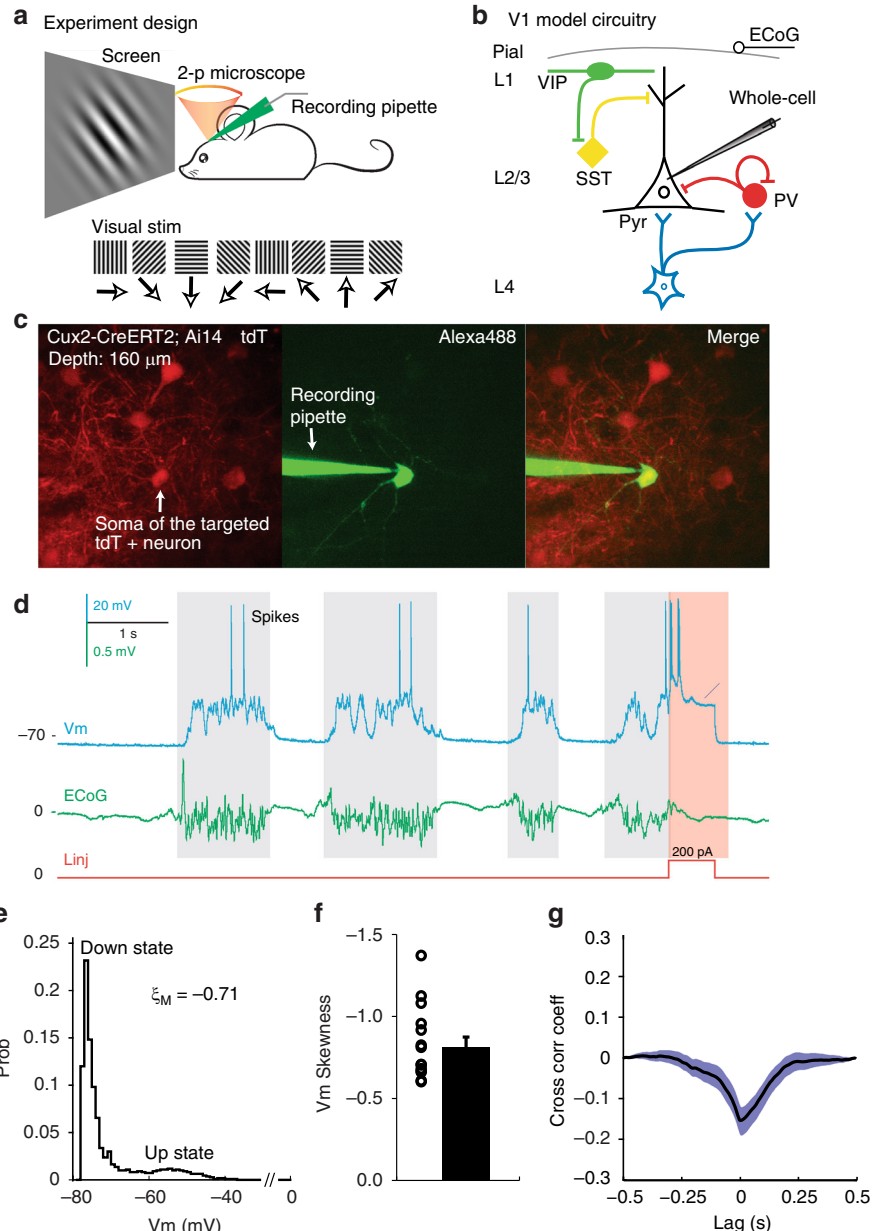

**Fig. 3** Excitatory pyramidal neurons couple strongly with local network dynamics. **a**, **b** Experiment design. **a** L2/3 & 4 tdT+ neurons in mouse V1 were visualised with 2-p microscopy and subjected to targeted whole-cell recordings under anaesthesia. In a subset of animals, ECoG was simultaneously recorded (see Methods). **b** V1 model circuitry showing four major neuron types: Pyr neurons, SST, PV and VIP INs. **c**, **d** Example targeted whole-cell recording of a V1 L2/3 tdT+ Pyr neuron (Pyr#15) in an Isoflurane anaesthetised *Cux2-CreERT2;Ai14* mouse. **c** Z-projection images showing the targeted tdT + Pyr neuron (left) filled with green fluorescence dye Alexa488 (middle) after whole-cell recording was achieved. The merge view (right) validated the targeted cell (yellow). Recording depth was 160 μm. **d** Spontaneous Vm data of the Pyr neuron in **c** and simultaneously recorded ECoG. Note the spontaneous Vm oscillation between the Up (gray shade regions) and Down (the rest) state and faithful correspondence between the Up state of Vm and desynchronised phase of ECoG. High spontaneous Vm-ECoG correlation indicated a dominant network influence on this Pyr neuron under our recording conditions. As a result, brief electrical perturbation by a current step (red shaded region) failed to override the network influence. **e** Vm skewness $\xi_M$ of the Pyr neuron shown in **c**, **d**, note the skewed, bimodal distribution of Vm. **f** Population data of Vm skewness $\xi_M$ of 15 Pyr neurons. **g** Spontaneous Vm-ECoG cross-correlation of the Pyr neuron shown in **c**, **d** Shade area represented s.e.m. Error bar: s.e.m. The mouse image in a was reproduced from Li et al. 2017 Nat Commun. 8:15604. doi: 10.1038/ncomms15604 with permission

current steps failed to evoke spontaneous Up-Down Vm oscillations (Supplementary Fig. 6), together arguing against the possibility of latent but suppressed local network drive by the proximity of Vrest to Vrev. These Subtype I neurons thus were little affected by local network dynamics and indeed network-uncoupled soloists. Due to the lack of specific cellular markers for Subtype I SST neurons, no optogenetic manipulations could be done without undesired confounding effects, but depolarising Vm electrically by current steps, or through changes of anaesthesia depth failed to alter the unimodality of Vm distribution (Supplementary Fig. 6), suggesting this behaviour was likely an intrinsic property of Subtype I SST INs even when the brain network was artificially put under a highly synchronised state.

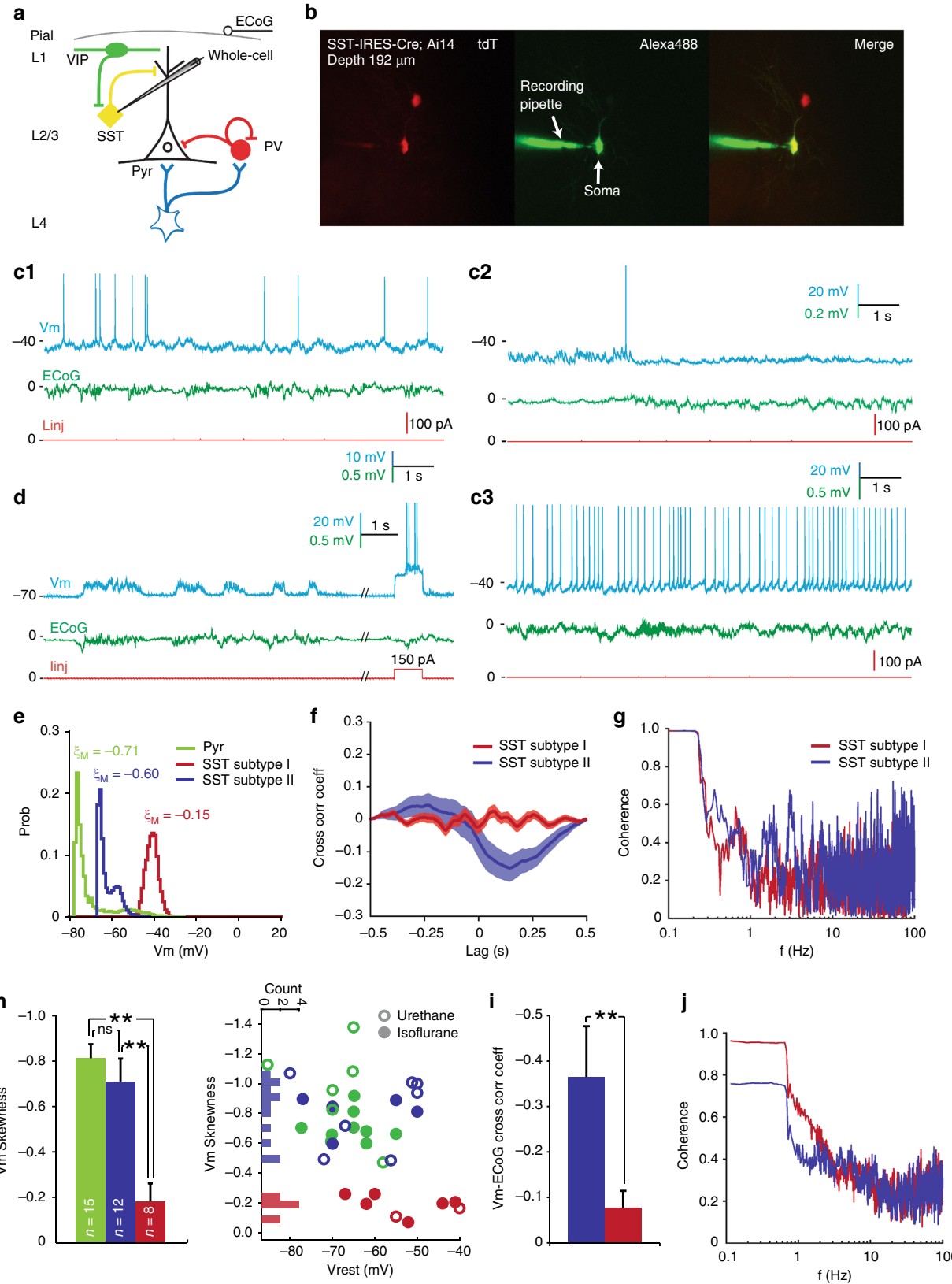

The rest of L2/3 SST neurons (i.e. Subtype II, 12 of 20, 60%), surprisingly, had bi-modally distributed Vm ($\xi_M = -0.71 \pm 0.10$, $n = 12$, $p < 0.01$ vs. Subtype I, $t$-test), which spontaneously oscillated between the Up and Down states (Fig. 4d, e and h). This resulted in a dual-peak distribution of $\xi_M$ in SST neurons $-(p < 0.05, n = 20$, bootstrapped Hartigan's Dip test), and well separated Subtype I from Subtype II (Fig. 4h). Vrest was hyperpolarised between $-50$ and $-80$ mV but not statistically different from Subtype I ($-62.3 \pm 3.3$ mV, $n = 12$, $p = 0.11$, Wilcoxon test, Fig. 4d, e and h, Supplementary Fig. 7). Vm-ECoG correlation was significantly stronger than that of Subtype I SST cells (CC $= -0.36 \pm 0.11$, $n = 4$, $p < 0.01$, $t$-test, Fig. 4f and i),

**Fig. 4** Targeted whole-cell recording confirms SST subtype specificity in neuron-network coupling. **a** 2-p targeted whole-cell recording in anaesthetised *SST-IRES-Cre;Ai14* mice. **b** Z-projection images of a L2/3 SST IN (Subtype I, SST#19) recorded at 192 μm. **c1**–**c3** Vm Unimodality and low Vm-ECoG cross-correlation persisted in SST Subtype I INs across Vrest and recording conditions. **c1** Spontaneous Vm and simultaneously recorded ECoG traces of a Subtype I SST IN (SST#17) under normal Isoflurane anaesthesia. Vrest was highly depolarised at ~−40 mV. Note the Up-Down brain states were prominent in ECoG but NOT Vm, indicating little network influence on this SST IN under our recording conditions. **c2**, **c3** Spontaneous Vm and ECoG traces of Subtype II SST#16 and #19, respectively. Cells were recorded at different depths of Isoflurane anaesthesia, but Vm unimodality and low Vm-ECoG coupling persisted. **d** Spontaneous Vm and ECoG traces of a Subtype II SST IN (SST#18) recorded at the same anaesthesia depth as **c1**. Vrest was at ~−70 mV. Note the good Vm-ECoG correspondence, suggesting dominant network influence. Recording depth: 188 μm. Neurons in **c1**, **c3** and **d** were from the same animal. **e** Vm skewness $\xi_M$ of **c1** (red) and **d** (blue) SST INs, note the different Vm distributions. The Pyr neuron (green) in Fig. 1e was re-plotted for comparison. **f** Spontaneous Vm-ECoG cross-correlation in **c1**, **d** . Shade area represented s.e.m. **g** Vm- ECoG coherence in **c1**, **d**. **h** Separation of Subtype I and II SST INs by Vm skewness $\xi_M$. Left: Population $\xi_M$ of Pyr, SST Subtype I and II neurons. Right: $\xi_M$ and Vrest in Pyr, Subtype I and II SST INs. SST Subtype I and II can be separated by $\xi_M$, and the separation was independent of Vrest and anaesthesia type. In some cells Vrest was slightly adjusted for display purposes. Open and solid symbols represented data acquired under urethane and Isoflurane anaesthesia, respectively. **i**, **j** Vm-ECoG cross-correlation and coherence in Subtype I and II SST neurons, respectively. Figure legends were the same for **e**–**j**. \*\*$p < 0.01$, n.s.: not significant. Error bar: s.e.m.

actually close to excitatory Pyr neurons (Fig. 4i), suggesting a strong local network inputs of excitation that intermittently brought Vm of Subtype II neurons to the Up state. The bi-modality of Vm distribution and tight Vm-ECoG correlation together demonstrated that Subtype II SST INs were strongly network-coupled. These electrophysiology data were highly consistent with the Ca imaging results in awake, head-fixed mice, indicating the SST subtyping was robust. It has been reported that some tdT+ cells in *SST-IRES-Cre;Ai14* mice could be PV INs[33,34], but we excluded this possibility from our dataset: firstly, none of the SST neurons included in our final analysis fired thin spikes (Figs. 3 and 4, and Supplementary Fig. 5). Secondly, the SST INs we analysed showed no characteristic PV behaviours, e.g. strong network synchronisation (Figs. 1 and 2). Thirdly, it has been confirmed that ~95% tdT+ cells[19] in V1 are indeed SST neurons in *SST-IRES-Cre;Ai14* mice. Lastly, to fully dismiss this possibility, we directly recorded PV neurons using *Slc32a-PV-Cre;Ai14* and *Gad2-IRES-Cre;Pvalb-2A;Ai14* mice. Our whole-cell data, consistent with previous studies in awake mice[8,32], showed PV neurons had a bi-modally distributed Vm (Supplementary Fig. 8). Our results thus together provided in vivo evidence that SST INs in mouse V1 L2/3 have at least two functionally distinct subtypes, although regional difference may exist between V1 and other cortical areas.

**SST subtype function predicted by network coupling profiles.** Different network coupling profiles predict different functional roles of SST neurons. For neurons tightly coupled with local networks, structured perturbations, such as sensory stimuli that produce large, simultaneous volleys of synaptic excitation will cause Vm transition to the Up state and desynchronisation to local networks[44]. This has been elegantly demonstrated in excitatory Pyr cells in rodents and monkeys, both during wakefulness and under anaesthesia[5,7,44]. However, for neurons that weakly couple with local networks, they would be much less impacted by sensory stimuli than strongly coupled ones. Our data showed it was exactly the case (Fig. 5). We managed to obtain reliable visually evoked Vm responses by stably recording SST neurons (n = 12) for ~15–60 min and compared their visually evoked Vm deflections (stimulus-evoked mean Vm difference after spike removal, see Methods) in response to whole-screen sinusoidal drifting gratings. Consistent with published data, Pyr neurons showed their characteristic responses with large Vm deflections (Fig. 5a and b). Subtype I SST INs, however, had significantly smaller Vm deflections to preferred gratings (3.6 ± 0.6 mV, n = 5, Fig. 5c and d), compared with that of Subtype II SST INs (14.0 ± 3.5 mV, n = 7, p < 0.01, Wilcoxon test, Fig. 5e–g). Importantly, Vm responses were highly consistent with the spontaneous Vm

oscillation pattern such that Subtype I SST INs can be completely separated from Subtype II by their Vm skewness and Vm deflection magnitude (Fig. 5g, Supplementary Fig. 9). The small Vm responses in Subtype I was not biased by Vrest (53.8 ± 3.5 mV vs. 63.4 ± 3.5 mV, p = 0.22, Wilcoxon test), because no strong inverse correlation[5] between Vm deflection and Vrest was found in Subtype I SST neurons (Fig. 5h), confirming Subtype I SST INs received little and non-specific sensory drive from the bottom-up visual pathway (Fig. 4i). These results verified Subtype I SST INs were network-uncoupled soloist neurons, and further suggested that they belong to the top-down feedback pathway, as demonstrated by the cross-modality modulation between V1 and A1 (ref. [45]. Our findings thus added new clarity towards the cell-type identity of soloist neurons and complemented previous results[8], because these regular-spiking SST neurons couldn't be distinguished from Pyr neurons by blind extracellular recordings. These results suggested that Vm activity contained critical information to study in vivo neuron-network coupling. Targeted cell-attached recordings in V1 L2/3 SST INs (n = 20) showed consistent results, suggesting this was not caused by recording artifacts of whole-cell patching (Supplementary Fig. 10). Taken together, our data provided the first piece of in vivo evidence that SST INs in mouse V1 L2/3 consist of two functionally distinct subtypes that may be involved in top-down and bottom-up pathways, respectively.

## Discussion
In-depth knowledge of spatiotemporally synchronised neuronal activity is essential to understand brain functions and dysfunctions, yet systematic understandings remain elusive. In the current study we combined 2-p Ca imaging with in vivo 2-p targeted electrophysiology and presented highly consistent results showing novel cell-subtype specificity in neocortical synchronisation. Compared with previous studies[17], our data covered 4 major neuron types, especially SST and VIP INs, and found novel subtype diversity in SST INs. SST Subtype I INs were uncoupled with local network dynamics due to their unimodally distributed Vm. They also showed low responsiveness to visual stimuli but could be highly spontaneously active[32] and sensitive to global brain states[32,40], suggesting a relation to the top-down pathway. SST Subtype II INs were strongly coupled with the network, illustrated by their tight Vm coupling with local network dynamics, bimodally distributed Vm that oscillated between the Up and Down states and higher responsiveness to visual stimulations, thus were more involved in the bottom-up pathway. Our results thus added a functional perspective to neuronal cell-typing. We also further dissected the cell-type identity of network-uncoupled soloist neurons, which can't be achieved with

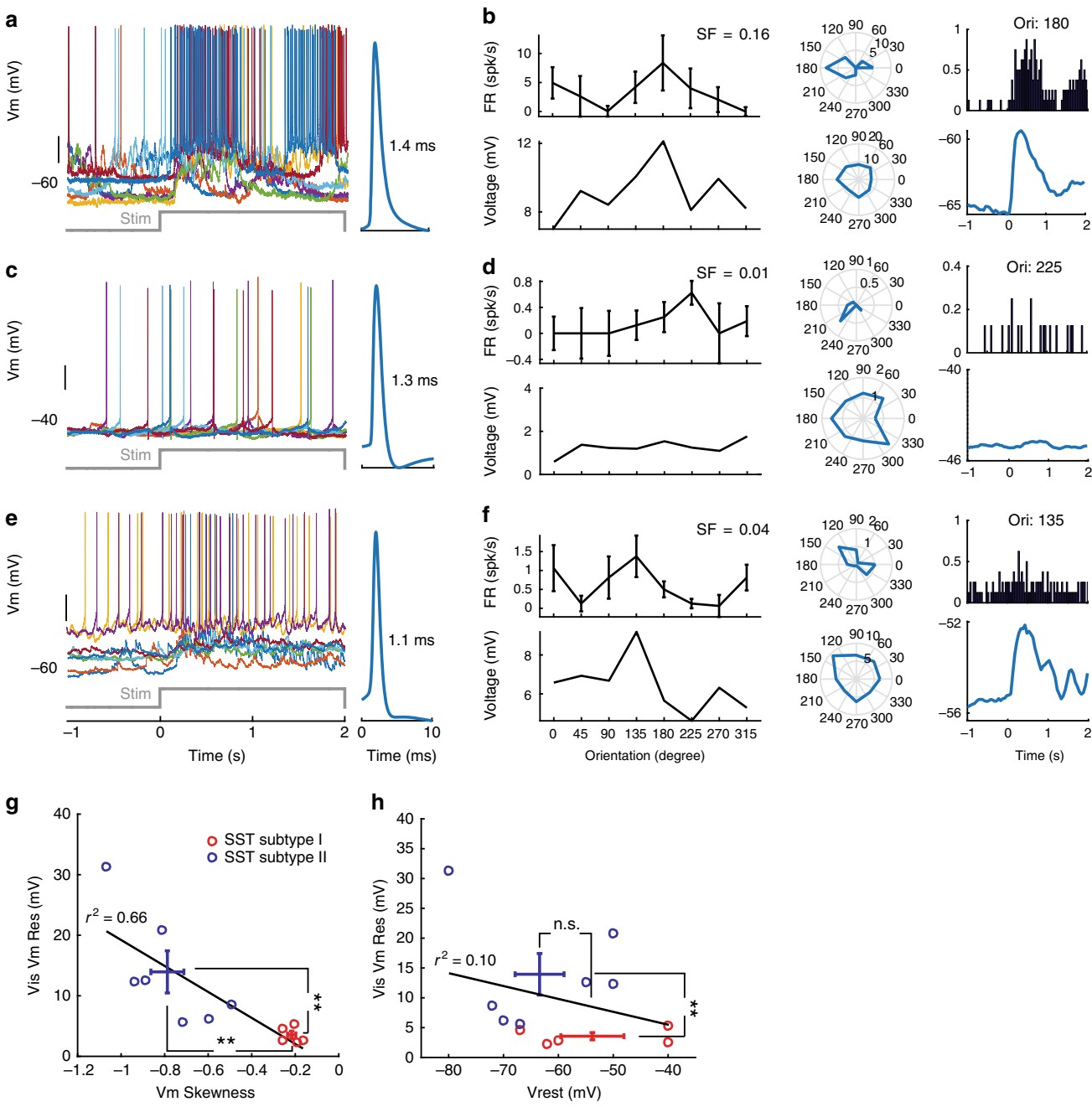

**Fig. 5** SST subtypes may play different functional roles. **a–f** Visual responses of example Pyr (**a**, **b**), SST Subtype I (**c**, **d**) and II (**e**, **f**) neurons. **a** Left: Vm traces of a *Cux2* Pyr neuron in response to its preferred drifting grating stimuli (eight repetitions). Time 0 marked the stimulus onset. Right: Average spike shape with FWHM value. Note that visual stimuli brought Vm to the Up state. Due to the spontaneous Up-Down Vm oscillation, the magnitude of Vm deflections depended on the pre-stim Vrest. **b** Spiking (top row) and Vm (bottom row) responses of the *Cux2* cell shown in **a**. Left and Middle columns: orientation tuning curves of spiking (top) and Vm (bottom) responses at the preferred spatial frequency (SF). Right column: Peri-stimulus time histogram (PSTH) of spiking responses to the preferred grating stimuli (top) and grand average of Vm responses (bottom). Note that Vm response was more broadly tuned than spiking response. **c**, **d** Visual responses of an SST Subtype I INs. Note the unimodally distributed Vm. Subtype I usually had low responsiveness to visual stimuli but could be highly spontaneously active. **e**, **f** Visual responses of an SST Subtype II INs. Note the bimodally distributed Vm and higher visual responses compared to Subtype I. **g** Properties of visually evoked Vm deflections confirmed the classification of SST subtypes by Vm dynamics and Vm-ECoG coupling. Visually evoked Vm deflection magnitude (Vis Vm Res) was plotted against Vm Skewness $\xi_M$. Subtype I and II SST INs were separated as expected. **h** No apparent dependence of Vm deflection magnitude on Vrest. **p** < 0.01, n.s.: not significant. Error bar: s.e.m.

blind extracellular recordings[8]. More importantly, our results might lead to a strategy to parse in vivo coordinated neural dynamics through the neuronal cell-type/subtype specificity, which could facilitate the simplification of neural dynamics that reverberate on brain circuits that consist of various types of neurons. Cell-type dependence has also been demonstrated

important to brain circuit re-organisation during development, learning and plasticity[46–50], and cell-type/subtype dependent mechanisms in normal brain also predict their key roles in brain dysfunction[13]. SST neurons provide feedback inhibition, the two functional SST subtypes demonstrated here may play significant roles in brain states, perception and cognition as suggested by

both experimental and theoretical works. It will be interesting to investigate whether and how subtypes of SST neurons exist in other cortical layers[51], or whether functional subtyping can be generalised for other neuron types, e.g. Pyr and VIP neurons. Future studies are needed to reveal cellular signatures of SST subtypes to allow targeted, larger-scale characterisation of their anatomical, functional and genetic properties, and more importantly, optogenetic and/or pharmacological manipulations in order to further elucidate their roles in sensory information processing, development, learning and memory, and brain disorders.

## Methods

**Materials**. Experimental procedures were in accordance with NIH guidelines and approved by the Institutional Animal Care and Use Committees (IACUC) of Allen Institute for Brain Science and Sun Yet-sen University.

2-p Ca imaging were performed in adult transgenic mice (2–7 months, both sex, $n = 14$), including *Emx1-IRES-Cre;CaMK2a-tTA;Ai94* (GCaMP6s, $n = 3$), *Pvalb-IRES-Cre;Ai163* (GCaMP6s, n = 3), *SST-IRES-Cre;Ai14;Ai148* (GCaMP6f, $n = 4$), *VIP-IRES-Cre;Ai14;Ai148* (GCaMP6f, $n = 3$) and *Cux2-CreERT2;Ai14;Ai148* (GCaMP6f, $n = 1$). 2-p targeted whole-cell and cell-attached recordings were conducted in adult transgenic mice (2–6 months, both sex, $n = 71$) including *Cux2-CreERT2;Ai14* (dense L2/3 labeling, $n = 9$), *Rorb-IRES-Cre;Ai14* (dense labeling in L4 and lower L2/3, $n = 18$), *Scnn1a-IRES-Cre;Ai14* (dense L4 labeling, $n = 4$), *Slc32a-PV-Cre;Ai14* ($n = 6$), *Gad2-IRES-Cre;Pvalb-2A;Ai14* ($n = 2$), *SST-IRES-Cre; Ai14* ($n = 29$) and *VIP-IRES-Cre;Ai14* mice ($n = 3$).

**In vivo 2-p Ca imaging**. Detailed imaging procedures have been published previously[23]. Briefly, adult mice were implanted with a custom-made, pre-notched metal head-post and a craniotomy was performed over the left visual cortical area under anaesthesia. To improve imaging quality while retaining access for electrophysiology, silastic sealant (Kwik Sil) was applied to the craniotomy[52,53] before a ~3 mm square polycarbonate coverslip was placed over the exposed cortical area centering on 1.3 mm anterior and 3.1 mm lateral to the Lambda. Notches on the head-post, which were carefully pre-aligned during implantation, were used to position the coverslip and later locate V1 for Ca imaging. After 7 days of recovery from surgery, mice were habituated to head fixation and presentation of visual stimulations on an in-house made running device (a freely-rotating running disc) for 1 week before imaging experiments started. During imaging, mice were awake, head-fixed but allowed to run or rest on the freely-rotating running disc with or without visual stimuli presented on a calibrated LCD screen.

2-p Ca imaging was performed using a Bruker (Prairie) 2-p microscope with eight kHz resonant scanners, coupled with a Chameleon Ultra II Ti:Sapphire femtosecond laser system (Coherent). In the current study, we used *Emx1-IRES-Cre;CaMK2a-tTA;A94*, which expressed the GECI GCaMP6s in pan-excitatory cortical neurons, to characterise Pyr neurons in V1 L2/3. Only those Ai94 mice that never showed behavioural signs for epileptic brain activity[54] were included in final analysis. For inhibitory interneurons, we chose the Ai148 and Ai163 reporter lines[17] to co-express GCaMP6f or GCaMP6s together with a red, Calcium insensitive cytosol fluorescence protein tdT in PV, VIP and SST positive inhibitory interneurons[16], respectively. Co-expression of tdT with GECI facilitated imaging and following data analysis processes in those INs, because we found that due to variations in cell activity and/or GECI expression profiles, cell density of observable INs could be low. Fluorescence was excited at 920 nm wavelength with <70 mW laser power measured after objective and collected in two spectral channels using green (510/42 nm) and red (641/75 nm) emission filters. Imaging was conducted between ~150–300 μm underneath the pial surface (Supplementary Fig. 1A). Fluorescence images were acquired at the frame rate of 512 × 512 pixels, 30 Hz through a ×16 water-immersion objective lens (Nikon, NA 0.8), with or without visual stimulations.

In literature neuronal synchronisation has been primarily studied with electrophysiological techniques, such as EEG, LFP, extracellular and intracellular recordings[1–3]. Ca imaging, compared with electrophysiology, offers advantages of simultaneously monitoring activities of local clusters of single neurons under more natural and functional relevant conditions (e.g. awake, behaving mice). Combined with our recent advances in transgenic mouse tools[22,23], we fully exploited these advantages to cell-type specifically characterise the pair-wise, within-type correlation profiles of local neuron populations in awake, head-fixed mice. In the present study, correlation profiles were found dependent on neuronal cell-types, and novel cell-subtype specificity was revealed in neuronal synchronisation in awake brain. These results demonstrated the novel subtlety among neuron types/ subtypes in which they orchestrated with local network dynamics, and further suggested a promising strategy to decompose coordinated neuronal dynamics through the subtype-distinct behaviours.

With these being said, we recognised limitations of Ca imaging: firstly, due to the sensitivity issue, neurons with little or no spiking activity, or highly active but showing little firing rate change (e.g. some Subtype I SST neurons), or having

strong Ca buffering capacity (e.g. PV neurons) could be found fluorescently inactive thus couldn't be well distinguished by Ca imaging; secondly, even in fluorescently active neurons, spikes that failed to evoke large enough fluorescence change might escape from detection; thirdly and lastly, the fluorescence signal of Ca imaging reflected changes of intracellular Ca concentration, which may not be 100% associated with the spiking activity. To complement the Ca imaging data, we performed in vivo 2-p targeted electrophysiology (see below), due to its best spatiotemporal resolution, to corroborate our findings. We found results from both approaches highly consistent. Specifically in SST INs, although Ca imaging in awake *SST-IRES-Cre;Ai14;Ai148* mice might only have reported subsets of Subtype I and II SST neurons compared with our targeted whole-cell recordings, Ca imaging data highly matched with those from whole-cell recordings, which we found very convincing.

**2-p targeted electrophysiology**. Detail of experimental procedures has been published previously[55,56]. Briefly, mice were anesthetised with urethane (1.5 g/kg, 30% aqueous solution, i.p.) or Isoflurane (0.75–1.5% in $O_2$) and maintained at the adequate depth of anaesthesia[57]. A metal head-post was implanted and a circular craniotomy (~3 mm in diameter) was performed using skull thinning over V1 centering on 1.25 mm anterior and 2.25 mm lateral to the Lambda. Durotomy was conducted to expose V1 regions of interest that were free of major blood vessels to facilitate the penetration of recording micropipettes. During surgery the craniotomy was filled with Artificial Cerebrospinal Fluid (ACSF) containing (in mM): NaCl 126, KCl 2.5, NaH$_2$PO$_4$ 1.25, MgCl$_2$ 1, NaHCO$_3$ 26, glucose 10, CaCl$_2$ 2, in ddH$_2$O; 290 mOsm; pH 7.3 with NaOH to keep the exposed V1 region from overheating and/or drying. A thin layer of low melting-point agarose (1–1.3% in ACSF, Sigma-Aldrich) was applied to the craniotomy to reduce brain motion. The mouse body temperature was maintained at 37 ºC with a feedback controlled animal heating pad (Harvard Apparatus).

2-p targeted electrophysiology was performed either manually[32,42,55,58] or assisted with a robot smartACT[56] on randomly chosen neurons with apparent tdT fluorescence. Long-shank borosilicate (KG-33, King Precision Glass) micropipettes (5–10 MΩ) were pulled with a P-97 puller (Sutter) and installed on a MultiClamp 700B headstage (Molecular Devices), which was mounted onto an MP-285 4-axis manipulator (Sutter) with an approaching angle of 31° from the horizontal plane. For whole-cell recordings, micropipettes were filled with K-Gluconate based internal solution containing (in mM): potassium gluconate 125, NaCl 10, HEPES 20, Mg-ATP 3, Na-GTP 0.4, in ddH$_2$O; 290 mOsm; pH 7.3 with KOH; Alexa488, 50 μg/ml. Individual tdT+ neurons within 100–350 μm underneath the pia surface were visualised with an Sutter Moveable Objective Microscope (Sutter) coupled with a tunable Chameleon Ultra II Ti:Sapphire femtosecond laser system (Coherent), controlled by the open-source software ScanImage 3.8 (Janelia Farm Research Campus/Vidrio Technologies). Fluorescence excited at 920 nm excitation wavelength with <70 mW laser power measured after the objective lens (×40 water-immersion, Olympus LUMPLFLN 40XW, NA 0.8) was collected in two spectral channels using red (641/75 nm) and green (510/42 nm) emission filters (Semrock) to visualise the target tdT+ neuron and Alexa488-containing micropipette, respectively.

Neurons were patched with standard techniques[32,42,56,58] to form the Giga-seal (g-seal, >1 GΩ, typically 2–8 GΩ), and whole-cell configuration was achieved after g-seal formation by rupturing the membrane (break-in) with brief applications of gentle negative pressure. $V_{rest}$ was measured immediately after break-in. Series resistance ($R_{series}$) was compensated, and input resistance ($R_{in}$) and access resistance ($R_a$) were measured. Recordings typically started ~2–5 min after break-in to allow sufficient diffusion of the internal solution. Vm data were acquired under current clamp mode with a Multiclamp 700B, digitised with a Digidata 1440A at 20 kHz, recorded by pClamp software (Molecular Devices) and stored on a PC (Dell). We did not correct liquid junction potential. In a subset of cells (12 Pyr, 13 SST and three PV, respectively), we recorded Vm responses to step, ramp, sinusoidal or white noise current injections (various amplitude, 500 ms in duration) for intrinsic membrane properties. For cell-attached recordings, the pipette was filled with ACSF and Alexa488, and data were acquired under "I = 0" mode (zero current injection). During recording, anaesthesia depth was regularly monitored by breath rate, absence of corneal/hindlimb reflex or spontaneous whisker/limb movements and maintained at appropriate depth[57] (Stage III-3 unless specified) as stable as possible by supplementing Urethane (10% of the original dose, i.p.) or adjusting the Isoflurane concentration. Experiments were terminated ≤14 h after the induction of anaesthesia, to ensure data quality.

**ECoG recording**. In a subset of *Cux2-IRES-CreERT2;Ai14* (n = 2) and *SST-IRES-Cre;Ai14* (n = 9) mice, two stainless steel wires were implanted over the ipsilateral prefrontal cortex (~1.0 mm anterior and ~1.0 mm lateral to the Bregma) and exposed V1 (~0.5–1 mm away from the duratomy), respectively, for ECoG recording. To minimally disturb following whole-cell recordings, the end of the wire in V1 was placed over the dura surface instead of being inserted into the cortex. This offered us recording stability with an improved spatiotemporal resolution of local network dynamics comparable to LFP, as shown by our data (Figs. 3 and 4). ECoG signal was amplified with a differential amplifier (DAM-50, World Precision Instruments) at 1000x gain, low-passed filtered at 200 Hz, digitised with a Digidata 1440A at 20 kHz, acquired using pClamp software and stored on the same PC (Dell).

**Visual stimulations**. Whole-screen sinusoidal drifting gratings were selected due to the reported size-tuning of SST INs[59]. Stimuli were presented with the calibrated LCD monitor spanning 60° in elevation and 130° in azimuth to the contralateral eye showing eight orientations (45° increment), 3 or 6 spatial frequency ([0.02, 0.04 and 0.08] or [0.01, 0.02, 0.04, 0.08, 0.16 and 0.32] cycle per degree, cpd) and 1 temporal frequency (2 Hz), at 80% contrast in a random sequence with 5–8 repetitions. To facilitate in vivo whole-cell data acquisition, each drifting grating lasted for 2 s with an interstimulus-interval of 2 s. A gray screen at mean illuminance was presented randomly for 16 times. The mouse's eye was positioned ~22 cm away from the centre of the monitor. Whole-screen stimuli Due to the choice of whole-screen visual stimuli, we didn't map the receptive field or tract the eye position.

**Data analysis**. Ca imaging and electrophysiological data were analysed using in-house Matlab scripts. For Ca imaging, image stacks were first corrected for in-plane motion artifacts using cross-correlation motion correction method between frames[60]. Image segmentation was done using Independent Component Analysis (ICA) followed by automated ROI selection[61] with manual confirmation/correction. We manually selected FOVs with typically ≥~15 well-identified ROIs for further processing. Green fluorescence signal of all pixels within individual ROI was averaged by image frames. To calculate spontaneous ΔF/F, the baseline for ΔF/F calculation was determined by the mode of spontaneous F trace. We then determined fluorescently active ROIs by thresholding the ΔF/F traces at five standard deviations, followed by manual verifications. We did not quantitatively analyse the fluorescence signal from the red channel.

For spontaneous within-type correlations, CCs were computed between ΔF/F traces of fluorescently active neuron pairs within the same FOV. Population data of CC between active neuron pairs were reported in Fig. 2e–h. To obtain CC profiles of individual neurons, zero-time lag CC was calculated between each active neuron and its active neighbours, and compared against a threshold value (see below). If an active neuron pair exhibited subthreshold zero-time lag CC, this pair of neurons was scored as uncorrelated; otherwise it was scored as correlated. For each active neuron, we then quantified the within-type correlativity by calculating the percentage of its correlated active neighbours within the same FOV.

To determine the CC threshold, we examined the SST data, which showed clear signs for subtypes. We first took the mean CC curves of individual SST neurons by averaging CC curves across all active SST neighbours in the same FOV. Then a sliding t-test was conducted on the distribution of mean zero-time lag CC (Supplementary Fig. 2A). Due to the good separation between Subtype I and II, typically the threshold showed a wide range. We then took the mid-point value, which was 0.15, as shown by the population data (Supplementary Fig. 2B), and used it as the empirical threshold for all cell types. The choice of CC threshold was validated by the high consistency of results in Pyr neurons between the present and previous studies.

A recent study reported aberrant epileptiform activity in some *Emx1-IRES-Cre; CaMK2a-tTA;Ai94* mice[54], which was expected to induce strong synchronisation thus biased our CC results. In the present study, special care was taken to our Ai94 dataset. No seizure-like behavioural signs were noticed in Ai94 mice included in our final data analysis, which was consistent with our Ca imaging data of normal cortical dynamics (e.g. Fig. 1d). Overall Pyr correlativity in Ai94 mice was found weak and fully comparable with that previously reported in normal mice[8,14–17], ruling out possible confounding influences by epileptiform cortical activity that would have dominated the CC profile. Taken together, available data suggested our Ai94 results were NOT contaminated by epileptiform activity. In summary, our imaging results were highly consistent with previous studies[14–17], validating the Ca imaging approach.

To control the known diverse effects of locomotion on activities of different neuron types[38,39], in our experiments mice were NOT required to continuously run on the running disc. As quantified in Supplementary Fig. 2, on average locomotion only occurred during ~5% −10% of total imaging time, we thus concluded that the within-type CC was largely measured during the stationary state. To investigate the effects of locomotion on within-type CC, velocity data were smoothed by 1 s time window then thresholded at 1 cm per sec (ref.[35]), locomotion CC was calculated between neuron pairs using ΔF/F segments when locomotion lasted for >2 s.

For visually evoked Ca responses, we took the baseline F as the average of the fluorescence signal within 1 s immediately before the stimulus onset, and ΔF/F was calculated for individual trials. To calculate the orientation selectivity index (OSI), ΔF/F was averaged over the 2 s stimulus duration, then by stimulus conditions. The preferred orientation ($\theta_{pref}$) was then determined as the orientation that evoked the strongest responses at the preferred SF. And OSI was calculated as $(R_{pref} - R_{orth})/(R_{pref} + R_{orth})$, where $R_{pref}$ and $R_{orth}$ stands for the response magnitude at $\theta_{pref}$ and the orthogonal orientation ($\theta_{pref} + \pi/2$), respectively. Response CC was computed between active neuron pairs using the ΔF/F traces averaged across stimulus conditions. We did not analyse the off response. To estimate how SST Subtype I neurons spatially distributed within the x–y plane, we compared the distribution of the distance between centres of pairs of identified SST Subtype I ROIs with that of dots that uniformly distributed within a two-dimension (2-d) square at the same size of the FOV. Our rationale here is: if the distance distribution of centroids we observed from our data approximates the theoretical distribution, it would provide

strong evidence that SST Subtype I neurons distributed evenly within the focal plane, and vice versa.

We first derived the theoretical distribution. For random variable (x, y) that distributes evenly in a 2-d square with length of a, its joint distribution density function f(x, y) is:

$$f(x, y) = 1; \qquad (1)$$

The Euclidean distance d between a pair of dots $(X_1, Y_1)$ and $(X_2, Y_2)$ can be written as:

$$d = \sqrt{(X_1 - X_2)^2 + (Y_1 - Y_2)^2}; \qquad (2)$$

It can be mathematically deducted[62] that g(d), the probability density function of d, is:

$$g(d) = 2d \times \begin{cases} -4\frac{d}{a^3} + \frac{\pi}{a^2} + \frac{d^2}{a^4}, & 0 < d \le a; \\ -\frac{2}{a^2} + \frac{4}{a^2}\sin^{-1}\left(\frac{a}{d}\right) + \frac{4}{a^3}\sqrt{d^2 - a^2} - \frac{\pi}{a^2} - \frac{d^2}{a^4}, & a < d \le \sqrt{2}a; \end{cases} \qquad (3)$$

We then plotted g(d), as shown in Supplementary Fig. 2C, when we set a = 600 μm, which was the maximal (diagonal) distance in our experimental dataset. It was then compared with the histograms of the Euclidian distance between centres of SST Subtype I and II ROIs. Compared with the theoretical distribution, the empirical histogram of Subtype II showed a left shift, suggesting Subtype II cells tended to cluster with its close neighbours, thus we concluded SST Subtype I distributed more evenly within the x–y plane, but Subtype II were more clustered, especially within 200 μm.

For electrophysiological data, to dismiss possible confounding effects by anaesthesia type, we first compared the Vm data between Isoflurane and Urethane anaesthesia. No apparent biases were found in distributions of Vrest (Isoflurane vs. Urethane, Subtype I: −54.3 ± 4.7 vs. −47.5 ± 6.1, p = 0.45, t-test; Subtype II: −63.4 ± 5.6 vs. −61.6 ± 4.4, p = 0.80, t-test), or the skewness of spontaneous Vm (Isoflurane vs. Urethane, Subtype I: −0.20 ± 0.03 vs. −0.14 ± 0.03, p = 0.30, t-test; Subtype II: −0.77 ± 0.05 vs. −0.80 ± 0.09, p = 0.82, t-test; same results from Wilcoxon test) across mice anesthetised by Isoflurane or Urethane (Fig. 4h). Thus we pooled the data acquired under urethane and Isoflurane anaesthesia together due to the exhibited consistency. For whole-cell recordings, cells showing Ra >60 MΩ or no apparent spike overshoot were excluded from final analysis. To exclude damaged/unhealthy cells, only those recorded cells remained stable during the entire recording session (~15 min to >1 h) were included in our final analysis. To characterise the subthreshold Vm activity, Vm data were detrended at 0.01 Hz, low-pass filtered at 2 kHz and spikes were removed by median filtering Vm with a 5–8-ms sliding window[63]. In order to compare the distributions of subthreshold Vm between Pyr, PV, SST Subtype I and I neurons, we calculated the Pearson's Coefficient of Skewness $\xi_M$ from the mode $M_{Vm}$, mean $\mu_{Vm}$ and standard deviation s of Vm, given the fact that Vm showed bimodal distributions in many neurons[64,65]:

$$\xi_M = (M_{Vm} - \mu_{Vm})/s; \qquad (4)$$

To verify the results, we calculated an alternative measure of Vm skewness $\xi$ as the third standardised moment of Vm, using $M_{Vm}$ to estimate the central tendency[66] due to the exhibited multimodal distribution of Vm:

$$\xi = E((Vm - M_{Vm})^3/s^3); \qquad (5)$$

Alternative to the median filtering of Vm, Vm was detrended at 0.01 Hz, low-pass filtered at 2 kHz, then thresholded at −20 mV to detect spikes, and Vm data within −5 to 5 ms around detected spikes were removed[7]. Consistent results were found between these two methods (Supplementary Fig. 5).

For Vm-ECoG cross-correlation, Vm and ECoG were first normalised by subtracting the mean then divided by the standard deviation, respectively. To obtain error bars, the spontaneous recording was divided into 15-s long segments, and the Vm-ECoG cross-correlation was calculated. In Pyr neurons Vm-ECoG correlation typically peaked at 0 ms, so the correlation at zero-time lag (mean from −2 to 2 ms across segments) was reported[5,7]. SST neurons however, showed delayed Vm activity (Fig. 4d and f) probably due to their facilitatory synaptic responses[67]. Thus the peak correlation value within −0.2–0.2 s time lag was reported for SST neurons to compensate for the shift of the correlation peak in time. We also computed the cross-correlation from the entire spontaneous recordings, and the results were similar[68]. Vm-ECoG coherence was estimated within 0.01–100 Hz using Chronux package (http://chronux.org). For cell-attached recordings, Vm data were thresholded at >4 standard deviation of Vm for spike detection.

To compare visually evoked responses, spikes during baseline recording (1 s pre-stimulus) and 0.2–2 s post-stimulus time were counted and averaged by stimulus conditions. Firing rate change between the post- and pre-stimulus time was then reported. Global orientation selectivity index (gOSI) was calculated as described before[69]. We didn't analyse the off response. For subthreshold Vm responses, spikes were removed then we averaged Vm response by stimulus conditions to minimise the impact of spiking and spontaneous Vm oscillation, and Vm deflections were obtained as the difference between the maximal post-stimulus and mean pre-stimulus of the average Vm response.

**Reporting summary**. Further information on research design is available in the Nature Research Reporting Summary linked to this article.

## Data availability

The authors declare that all data supporting the findings of this investigation are available within the article, its Supplementary Information, and from the corresponding authors upon reasonable request.

## Code availability

Custom code used for data analysis is available upon request.

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

## Acknowledgements

We thank Brian Long, Adrian Cheng, Kenji Mizuseki, Natalia Orlova, Elliot Mount, Hanchuan Peng and Christof Koch for the help. We wish to thank the Allen Institute founders, Paul G. Allen and Jody Allen, for their vision, encouragement and support. This work was also supported by grants from National Science Foundation of China 31871055 (L.L.), Guangdong Science and Technology Department Programs 2017B030314026 and 2018B030334001 (L.L.).

## Author contributions

L.L. conceived the project. L.L., U.K. and L.H. performed the experiments. L.L., U.K. and L.H. analysed the data. L.L. and H.Z. supervised the project. L.L wrote the paper with inputs from all authors.
