## [Peer Review File · Nature Communications]

Editorial Note: this manuscript has been previously reviewed at another journal that is not operating a transparent peer review scheme. This document only contains reviewer comments and rebuttal letters for versions considered at *Nature Communications*.

REVIEWERS' COMMENTS:

Reviewer #1 (Remarks to the Author):

This revised manuscript by Knoblich et al. has addressed most of my concerns. I have only a few remaining issues that need to be addressed before publication.

Specific points:

- 1) Does the subtype I SST neuron correspond to the SST neuron in Genet et al., 2012 study, which is also uncoupled from the network? The authors need to discuss about this possibility.
- 2) Line 135, "These results suggested SST Subtype I INs were more sensitive to locomotion..." but as compared with which group of neurons and by what criteria? By looking at the change of mean CC, pyramidal cells are more sensitive to locomotion than subtype I SST neurons.
- 3) In Fig. 2J, the labeling is in "fraction" but not in "percentage".

Reviewer #2 (Remarks to the Author):

The authors have carefully addressed the concern about anesthesia depth and difference between isoflurane and urethane, and I have no farther questions in this respect.

In their response the authors asked how anesthetized cortex can be more desynchronized than awake. Urethane is known to be able to produce a highly desynchronized state (e.g. see Fig. 4 in Renart et al. Science 2010), in awake rodents such state is achieved only in a highly activate conditions.

The authors have addressed the point about epileptic transgenic mice (however on page 15 the wrong paper seems to be cited).

In figs 3&4 I would specifically state what the shaded areas are.

Code availability (from Nature checklist) - I failed to spot any statement to this effect.

Reviewer #3 (Remarks to the Author):

Knoblich et al present a revised manuscript with largely the same data set. The paper is more rigorously presented, and they have addressed many of the other reviewers' concerns. My main critique from the initial review was that this study is exclusively phenomenal – it presents some interesting new data on cell-cell correlations of inhibitory neurons in mouse V1, with the most interesting finding being the sub-type of SST neurons that don't apparently couple to the network. This revision, however, adds neither any new mechanistic data to explain this lack of coupling, nor any insight into how different levels of coupling among the interneurons has any specific impact on V1. Thus the impact of this study is still going to be very limited. The data set will be of interest to a focused set of researchers specifically trying to understand mostly spontaneous correlations in the brain. The spontaneous correlations observed here probably do tell us something interesting about the connectivity of the V1 network – for instance, the hypothesis the authors raise that the uncoupled SST subtype probably doesn't receive neither bottom nor lateral input seems plausible (although in no way substantiated here). They propose they it instead received top down input specifically – also an interesting notion, but also not tested in any way. Taken together, my critique remains largely the same. The study seems more like a preamble to more interesting follow up paper.

We thank the reviewers of our manuscript again for their positive feedback, which we highly appreciate. We have revised our manuscript in response to the comments we received, as we summarized below. We also proposed a new title of “**Physiologically distinct subtypes of Somatostatin-expressing inhibitory interneurons in mouse primary visual cortex**”, to emphasize on the novelty of SST subtypes presented by our data.

Itemized Responses to Reviewer #1:

Specific point #1) Does the subtype I SST neurons correspond to the SST neuron in Genet et al., 2012 study, which is also uncoupled from the network? The authors need to discuss about this possibility.

Response: We noticed the similarity between the SST Subtype I neurons in the current study and those reported in Genet et al., 2012 in awake S1. This similarity was mentioned in our manuscript (**page 7, paragraph #2**). We think there is a high probability that the Subtype I in the current study largely corresponds with those S1 SST neurons in Genet et al., 2012, although a regional difference may exist, as we stated in our previous response letter.

Specific point #2) Line 135, “These results suggest SST Subtype I Ins were more sensitive to locomotion...” but as compared with which group of neurons and by what criteria? By looking at the change of mean CC, pyramidal cells are more sensitive to locomotion than subtype I SST neurons.

Response: We agree with the reviewer that this sentence introduced confusion. In the revised manuscript we clarified that the comparison was made between SST Subtype I and II neurons.

Specific point #3) in Fig. 2J, the labeling is in “fraction” but not in “percentage”.

Response: We apologize for this error and have it corrected in the revised manuscript.

Itemized Response to Reviewer 2:

Specific point #1) In their response the authors asked how anesthetized cortex can be more desynchronized than awake. Urethane is known to be able to produce a highly desynchronized state (e.g. see Fig. 4 in Renart et al. Science 2010), in awake rodents such state is achieved only in a highly activate conditions.

Response: We appreciate the input from the reviewer and are glad to know that there actually is no disagreement regarding this issue.

Specific point #2) The authors have addressed the point about epileptic transgenic mice (however on page 15 the wrong paper seems to be cited).

Response: We have corrected the reference on page 15 and feel quite happy to know that we have addressed the concern of possible epileptic activity in Ai94 mice.

Specific point #3) in Figs 3 & 4 I would specifically state what the shaded area are.

Response: We apologize for this and have it clarified in the revised manuscript.

Specific point #4) Code availability (from Nature checklist) – I failed to spot any statement to this effect.

Response: We thank the reviewer for pointing out this for us and have added the Code Availability statement to the revised manuscript.

Itemized Response to Reviewer 3:

Specific point #1) This revision, however, adds neither any new mechanistic data to explain this lack of coupling, nor any insight into how different levels of coupling among the interneurons has any specific impact on V1. Thus the impact of this study is still going to be very limited. The data set will be of interest to a focused set of researchers specifically trying to understand mostly spontaneous correlations in the brain. The spontaneous correlations observed here probably do tell us something interesting about the connectivity of the V1 network – for instance, the hypothesis the authors raise that the uncoupled SST subtype probably doesn't receive neither bottom nor lateral input seems plausible (although in no way substantiated here). They propose they it instead received top down input specifically – also an interesting notion, but also not tested in any way. Taken together, my critique remains largely the same. The study seems more like a preamble to more interesting follow up paper.

Response: We appreciate the comments by the reviewer, which we believe will only make our manuscript better. As we stated in our previous response letter, we think the current study presents novel findings to the exploration of sorting out how neurons types coordinate in the synchronized neural ensemble. We hope our results, once get published, would be constructive to other groups who feel interested in this intriguing scientific question. There is no doubt about the authors' enthusiasm towards elucidating the identity or functional roles of SST subtypes in cortical computation, as we are planning to conduct studies aiming at full-morphology reconstruction of functionally characterized single SST neurons. But as the old saying goes, Rome was not built in a day, the authors will feel happy if can help to put down some pieces of stone.